# The Nutrient Content, Growth, Yield, and Yield Attribute Traits of Rice (*Oryza sativa* L.) Genotypes as Influenced by Organic Fertilizer in Malaysia

Mohammad Anisuzzaman [1,2], Mohd Y. Rafii [3,*], Shairul Izan Ramlee [3], Noraini Md Jaafar [4], Mohammad Ferdous Ikbal [1,5] and Md Azadul Haque [1,5]

1   Laboratory of Climate-Smart Food Production, Institute of Tropical Agriculture and Food Security, Universiti Putra Malaysia (UPM), Serdang 43400, Malaysia; anis.breeding94@gmail.com (M.A.); binaikbal@gmail.com (M.F.I.); mahaquebina@gmail.com (M.A.H.)
2   Plant Breeding Division, Bangladesh Rice Research Institute (BRRI), Gazipur 1701, Bangladesh
3   Department of Crop Science, Faculty of Agriculture, Universiti Putra Malaysia (UPM), Serdang 43400, Malaysia; shairul@upm.edu.my
4   Department of Land Management, Faculty of Agriculture, Universiti Putra Malaysia (UPM), Serdang 43400, Malaysia; j_noraini@upm.edu.my
5   Bangladesh Institute of Nuclear Agriculture (BINA), Mymensingh 2202, Bangladesh
*   Correspondence: mrafii@upm.edu.my; Tel.: +601-9309-6876

**Abstract:** One of the most important challenges to continuously maximizing crop production on limited areas of agricultural land is maintaining or enhancing soil fertility. Organic fertilizer application is needed to replace nutrients recovered by crops from the fields in order to restore the crop production potential of the soil. The utilization of chicken manure as an organic fertilizer is essential in improving soil productivity and cop production. In Malaysia, demand for rice as a food source is rising in tandem with population growth, while paddy rice production capacity is becoming increasingly constrained. Field experiments were carried out in Sungai besar, Kuala Selangor, Malaysia during the two planting seasons in 2020 to evaluate the effects of different levels of organic fertilizer on the growth and yield of rice genotypes. A split plot layout in a randomized complete block design with three replicates was used. The twelve rice genotypes were in the main plots. The sub-plots were treatments. The experiment comprised 4 treatments, viz., $T_1$ = 100% NPK ($N_{150}P_{60}K_{60}$), $T_2$ = Chicken manure @ 5 t ha$^{-1}$, $T_3$ = Chicken manure @ 7 t ha$^{-1}$, and $T_4$ = Chicken manure @ 10 t ha$^{-1}$. The study indicated that different levels of chicken manure and NPK fertilizer showed significant effects on growth, yield, and yield contributing characters of all the rice genotypes. Results showed that application of chicken manure 10 t ha$^{-1}$ was the best in producing growth and yield contributing characters, grain and straw yields, and also nutrient (N, P, and K) contents in grain and straw. The maximum number of panicles (422.56 panicles m$^{-1}$), the maximum number of filled grains (224.49 grains panicle$^{-1}$), and the maximum grain yield (8.02 t ha$^{-1}$) and straw yield (9.88 t ha$^{-1}$) were recorded from $T_4$ treatment at the rice genotype BRRI dhan75. Although the highest biological yield was recorded from $T_4$ treatment, a statistically similar result was found for $T_3$ treatment. The highest harvest index was also recorded for $T_4$ treatment. Therefore, rice genotype BRRI dhan75 can be recommended under chicken manure @ 10 t ha$^{-1}$ for rice production in Malaysia.

**Keywords:** chicken manure; NPK; growth; yield component; nutrient content

## 1. Introduction

Rice is a staple cereal food in social nutrition and the primary food grain for more than a third of the world's population [1]. Asian countries produce and consume 90% of the world's rice. In rice farming, India and China account for half of the total area. Most countries' agriculture has advanced in recent years as a result of the green revolution,

which has included the adoption of high-yielding cultivars, improved irrigation systems, new farming practices, and fertilizers. Rice is a great source of carbs with roughly 87% of the grain being carbohydrates. It contains 7 to 8% protein, which is more digestible, biologically valuable, and nutritious, as well as less crude fiber and fat (1 to 2%). Rice alone provides around 20% of the world's dietary energy, which is more than maize or wheat [2].

Nitrogen (N) is the most important and critical plant nutrient, and it increases crop yields in a good way [3]. Nitrogen is required by all crops, regardless of soil type. Soil minerals, soil organic matter, rice straw, and manures are all natural sources of nitrogen. The application of organic matter such as manures can only replace soil organic matter in the short run [4]. To guarantee that the soil is fully supplied with all of the plant nutrients in an easily available form and to maintain good health and organic manures must be used in conjunction with inorganic and organic fertilizers to achieve optimal yields [5]. One theory is that improved yields can be achieved by combining current safe technology with traditional agriculture in an untested form.

In recent years, to increase rice yields, farmers have used a lot of artificial fertilizers, especially nitrogen fertilizers. The majority of farmers use fewer phosphorus and potassium fertilizers [6]. Rice yields are primarily increased by using significant volumes of artificial fertilizers. However, this has resulted in soil concerns, decreased food yields, and a slew of other environmental issues. As a result, we must create environmentally benign alternatives to chemical fertilizers that can supplement or replace them. Organic fertilizers are environmentally benign and can help to sustain soil health when used in concentrated rice production. They help to maintain the amount and quality of organic matter in the soil while also providing N, P, K, and important micronutrients [7,8]. For instance, using organic manure can result in significant increases in organic matter and plant accessible N, P, and K in the soil [9].

Organic manure can be a good source of plant nutrients, which can help increase crop yields [10]. Furthermore, this important component of soil is deteriorating with time due to exhaustive cropping and use of higher doses of chemical fertilizers with little or no accumulation of organic manure in the farmers' fields. Soil organic matter increases the physio-chemical properties of the soil and eventually promotes crop production [11]; so, it would not be prudent to depend only on the essential potential of soils for higher crop production. In more recent times, consideration has been given to the exploitation of organic wastes, farm yard manure, vermicompost, and poultry manure as the most effective measures for refining soil fertility and thus crop productivity [12]. The use of organic fertilizers needs to be applied for the enhancement of soil physical properties and of the supply of crucial plant nutrients for higher yields [13].

Furthermore, due to its high content of macronutrients, chicken manure has been chosen over other animal wastes [14]. For example, Dikinya and Mufwanzala [15] observed that, after applying manure to sandy soils and sandy loam soils, nitrogen levels increased by 40–60% and 17–38%, respectively, compared to control. Moreover, adding chicken excrement to the soil raises the concentration of water soluble salts. Plants receive their nutrition from soluble salts, but an excess of soluble salts (soil salinity) restricts plant growth. Gondek et al. [16] stated that chicken manure had an EC of roughly 11 dS/m in silt loam soils, which was too high for salinity sensitive crops. In most of the nutrients accessible in this environment, the pH of dry chicken dung pellets was determined to be 7.9 [14] while the availability of nutrients to plants was influenced by a reduction in soil pH (<6.5) [17,18] Furthermore, if properly applied, chicken manure can work as an effective soil amendment and/or fertilizer (e.g., providing N, P, and K), while also enhancing the N, P, K, Ca, and Mg concentrations in the soil and leaves [18]. These chemical features of soil provide information on chemical reactions on the processes affecting nutrient availability and the methods of replenishing nutrients in soils [19].

Many farmers are turning to organic or "low input" farming as a strategy for economic survival. Several comparisons of actual grain farms in Malaysia show that organic farming equals or exceeds conventional farming in economic performance. Organic farmers need

to borrow less money than conventional farmers for two reasons. First, organic farmers buy fewer inputs such as fertilizer and pesticides. Second, cost and income are more evenly distributed throughout the year on diversified organic farms. Organic farmers have complained that they are discriminated against by lenders, which is a possible economic disadvantage of organic farming. The relative economic performance of organic farming and conventional farming is sensitive to the ratio of input costs to the value of outputs. Both organic and conventional farmers are vulnerable to fluctuations in both input and output prices, but the effect of a given change will differ between the two farming system.

The future of commodity prices is not clear. However, changes in commodity prices can be expected to have greater impacts on conventional farmers than on organic farmers. Conventional producers have higher average yields for most grain crops. Therefore, assuming constant production costs, price increases will increase the net returns of conventional farmers by a greater proportion than those of organic farmers. Conversely, price decreases will decrease conventional returns by a greater proportion than organic returns. Differential price changes (increases in some commodity prices and decreases in others) would also tend to have effects of greater magnitude, whether positive or negative, on conventional farmers, since they depend on fewer crops for their income. Because organic systems are more diversified, the effects of differential price changes on income would partially offset each other. Increases in the cost of variable inputs would be less damaging to organic farmers because they purchase fewer inputs. The most likely price increases in the near future will be for energy, with consequent increases in the price of synthetic fertilizers. Organic farmers use less energy than conventional farmers, primarily because they use less synthetic fertilizer.

Rice production increasing through intensification by researchers but inorganic fertilizers generate several deleterious effects to the environmental and human health. Inorganic fertilizers should be replenished every cultivation season because the synthetic compounds of N, P, and K fertilizer are rapidly lost either by evaporation or by leaching in drainage water, and this leads to dangerous environmental pollution. Continuous usage of inorganic fertilizer affects soil structure. Organic manure decomposes to form humus which binds soil particles together, thus improving the soil structure and its physical properties.

Nitrogen (N), phosphorus (P), and potassium (K) are most limited nutrients in Malaysian soil due to various factors like acidity, continuous cropping, and continuous broadcasting of NPK where 60% is lost through volatilization. The increase in the use of inorganic fertilizer calls for a quick intervention because sustainable yield can be integrating by organic manure. Chicken manure should be preferred because it has most essential nutrients in high quantities and thus very little quantities are required to provide the essential nutrients. It is very light in weight and very rich in nutrients unlike other organic manures; this is why transportation is very easy. Its use in the long run improves nutrient holding capacity, and this lowers capital investment because much less inorganic fertilizer will be required.

The continuous use of inorganic fertilizers leads to deterioration in soil physical, chemical, and biological properties. It is true that sustainable production of crops cannot be maintained by using chemical fertilizers, but it is possible to obtain higher crop yields by using organic manure. Organic sources of nutrients are necessary for sustainable agriculture that can ensure food production with high quality. In intensive cropping, the continuous use of high levels of chemical fertilizers decreases crop productivity, as well as soil fertility status. The use of organic manure to meet the nutrient requirements of crops would be an invertible practice in years to come, particularly for resource poor farmers. Furthermore, ecological and environment concerns over the increased and indiscriminate uses of inorganic fertilizers have made research on uses of organic materials as sources of nutrients very necessary. More recently, attention has been given to the utilization of chicken manure as the most effective measure for improving soil fertility and thereby crop productivity. Application of chicken manure is needed for the improvement of soil physical properties and of the supply of essential plant nutrients for higher yields. Therefore, this

study was conducted to observe the effect of chicken manure on the growth and yield contributing characters of rice and to develop a suitable dose of chicken manure for rice.

Basically, the research presents scientific facts on the effect of chicken manure on overall rice production. The research developed appropriate recommendations for the chicken manure rates: 10 t/ha is to be applied by farmers, thus increasing rice production. Based on study, the results can act as a basis for advising farmers on use of rice that is high yielding and highly nutritive and matures within a short time. This research may end up as a source of knowledge on the effects of organic and inorganic fertilizer on overall crop production. This will also assist rice production using chicken manure.

## 2. Materials and Methods

### 2.1. Plant Materials

This study evaluated twelve rice genotypes. The genotypes came from various places. Table 1 presents the genotype names and their sources.

**Table 1.** List of the 12 rice genotypes used in the study.

| Code No | Name of Genotypes | Seed Sources |
|---------|-------------------|--------------|
| G1 | HUA565 | Philippine |
| G2 | MR297 | Malaysia |
| G3 | Putra1 | Malaysia |
| G4 | Putra2 | Malaysia |
| G5 | MR303 | Malaysia |
| G6 | MR309 | Malaysia |
| G7 | BR24 | Bangladesh |
| G8 | BRRI dhan48 | Bangladesh |
| G9 | BRRI dhan82 | Bangladesh |
| G10 | BRRI dhan72 | Bangladesh |
| G11 | BRRI dhan39 | Bangladesh |
| G12 | BRRI dhan75 | Bangladesh |

### 2.2. Experimental Location

The field experiment was conducted at Sungai Besar, Kuala Selangor, Malaysia, in a farmer's field. The experiment took place across two seasons, the first from September 2020 to January 2021 and the second from March to July 2021. Geographically, the location is around 3°38′ N latitude and 101°05′ E longitude, with an elevation of 3 m above sea level and a humid tropical environment. Table 2 contains the details of the weather information.

**Table 2.** Month-wise average of daily maximum temperature, minimum temperature, mean temperature, relative humidity, and rainfall at Sungai Besar, Kuala Selangor during experimentation period, 2020–2021.

| | 1st Planting | | | | | 2nd Planting | | | |
|---|---|---|---|---|---|---|---|---|---|
| | Temperature (°C) | | | Rain Fall (mm) | | Temperature (°C) | | | Rain Fall (mm) |
| Month | Max. | Min. | Ave. | | Month | Max. | Min. | Ave. | |
| August | 33.64 | 24.65 | 29.14 | 188 | February | 33.56 | 24.61 | 29.08 | 271 |
| September | 35.52 | 25.33 | 30.42 | 254 | March | 32.79 | 23.58 | 28.18 | 154 |
| October | 34.19 | 25.18 | 29.68 | 367 | April | 33.14 | 23.74 | 28.44 | 138 |
| November | 34.72 | 24.57 | 29.64 | 403 | May | 34.62 | 25.46 | 30.04 | 126 |
| December | 33.84 | 23.90 | 28.87 | 382 | June | 34.81 | 25.27 | 30.04 | 113 |
| Average | 34.38 | 24.72 | 29.55 | | Average | 33.78 | 24.53 | 29.15 | |
| Total | | | | 1594 | Total | | | | 802 |

## 2.3. Experimental Design and Treatment Combinations

The experiment consisted of four (4) treatment combinations with Chicken manure (CM) and the Chemical Fertilizer Recommended Rate (CFRR) for a high yield goal (HYG) as follows. $T_1$: 100% CFRR (150 N; 60 $P_2O_5$; 60 $K_2O$); $T_2$: 5 t/h CM; $T_3$: 7 t/h CM and $T_4$: 10 t/h CM. The experiment designed was laid out in a split plot design setup with three replications in a plot 32 m × 24.5 m, with a planting distance of 25 cm × 25 cm and a sub-plot size of 2 m × 1.5 m. Chicken manure was allocated to the main plot, while rice genotypes were assigned to the sub-plot.

## 2.4. Raising of Seedlings and Transplantation

The seeds of the twelve genotypes were first oven dried at 50 °C for 24 h to reduce pre-germination. To encourage germination, the seeds were immersed in water for 24 h. After 24 h, the seeds were withdrawn from the water and kept moist for 3 days in order to germinate. Water was added to each petri dish to avoid drying out and to keep it moist before it stats germinate. A soil-filled tray was provided for each genotype at the nursery, and soil with water was provided at optimum. After 3 days in the petri dishes, the seeds were transferred to the plastic filled tray in the nursery. Seeds were allowed to grow for 21 days, after which they were transplanted to the field.

## 2.5. Fertilizer Application and Intercultural Operations

Chicken manure was mixed into the soil before crop establishment and applied at rates of 5, 7, and 10 t ha$^{-1}$. Triple super phosphate and muriate of potash were sprayed during final plot preparation, and urea was administered in two split doses at 25 and 55 days after sowing (DAS) to supply the total recommended nutrient of 150 N:60 $P_2O_5$:60 $k_2O$ kg ha$^{-1}$. Organic and chemical fertilizers were used in accordance with the treatment recommendations. Weeding and other management activities were carried out as needed. Irrigation was also performed when needed.

## 2.6. Chemical Analysis of Soil and Chicken Manure Sample

Initial soil samples were gathered at a depth of 0–15 cm from the surface. After being freed of weeds, plant roots, stubble, and stones, the samples were dried and crushed to pass through a 2 mm (10 mesh) sieve. The samples were then placed in clean plastic bags for chemical and mechanical analysis. Standard procedures were used to evaluate the physical and chemical parameters of the initial and post-harvest soil samples in Table 3. The textural class was calculated by utilizing the USDA technique to project the percentages of sand, silt, and clay to the Marshall's triangular coordinate, and the hydrometer method was used to measure the particle sizes in the soil [20]. The Walkley and Black method was used to measure organic matter [21], a glass electrode pH meter was used to determine the soil pH (1:2.5 soil-water) [22], the semi-micro Kjeldahl method was used to calculate total nitrogen [23], the Olsen method was used to determine the available P [24], a flame photometer after extraction with 1N $NH_4OA_c$ at pH 7.0 was used to determine the exchangeable K [25], available S was obtained by extracting soil samples with $CaCl_2$ solution (0.15 percent) and measuring turbidity using a spectrophotometer method [26] method, and CEC was determined using the sodium saturation method [20].

**Table 3.** Physio-chemical parameters of the initial and post-harvest soil and manure samples in a two-season field experiment, 2020–2021.

| Soil Characters | Pre-Planting | After Crop Harvest | | | | | | | |
|---|---|---|---|---|---|---|---|---|---|
| | | 100% CFRR | | 5 t/ha CM | | 7 t/ha CM | | 10 t/ha CM | |
| | | 1st Planting | 2nd Planting | 1st Planting | 2nd Planting | 1st Planting | 2nd Planting | 1st Planting | 2nd Planting |
| pH | 6.5 | 6.43 | 6.48 | 6.65 | 6.73 | 6.79 | 6.82 | 6.91 | 7.02 |
| EC (µS/cm) | 63 | 69 | 77 | 65 | 69 | 68 | 73 | 71 | 75 |
| CEC | 17.68 | 19.36 | 18.44 | 18.62 | 18.58 | 18.79 | 19.27 | 19.53 | 20.08 |
| Organic carbon (%) | 0.77 | 0.85 | 0.89 | 0.87 | 0.93 | 0.91 | 0.96 | 0.93 | 1.04 |
| Organic matter (%) | 1.49 | 1.52 | 1.55 | 1.64 | 1.71 | 1.70 | 1.74 | 1.72 | 1.78 |
| Total N (%) | 0.08 | 0.19 | 0.22 | 0.10 | 0.13 | 0.12 | 0.14 | 0.13 | 0.16 |
| Exchangeable K (cmolkg$^{-1}$) | 0.31 | 0.39 | 0.43 | 0.32 | 0.33 | 0.33 | 0.34 | 0.35 | 0.37 |
| Available P (mgkg$^{-1}$) | 11.48 | 14.02 | 14.25 | 12.40 | 12.84 | 12.59 | 13.17 | 12.93 | 13.26 |
| Sand | 38.57 | 42.53 | 43.05 | 39.45 | 40.18 | 40.62 | 41.76 | 41.57 | 42.39 |
| Silt | 41.32 | 43.19 | 43.64 | 42.47 | 42.50 | 43.08 | 43.35 | 43.82 | 44.51 |
| Clay | 33.49 | 35.72 | 35.84 | 34.52 | 34.68 | 37.52 | 37.69 | 38.46 | 38.75 |
| Soil texture | Clay loam | Clay loam | Clay loam | Clay loam | Clay loam | Clay loam | Clay loam | Clay loam | Clay loam |

The pH was measured from the manure samples by a digital pH meter (HI 2211 pH meter, Hana instrument, Woonsocket, RI, USA) at the ratio of 1:10 ($w/v$) water-soluble extract, while a 1:20 ($w/v$) ratio of the sample with water was used to determine EC by digital EC meter. The total C, N, and S were determined by a Leco TruMac CNS analyzer. The C:N ratio was calculated from the total C and total N values. The TOC and OM contents of the manure samples were estimated by the loss on ignition method. The CEC was determined by the ammonium acetate (pH 7.0) leaching method. Inorganic N was determined using the method of Keeney and Nelson. The total contents of K, Ca, Mg, Na, Mn, Cu, and Zn were determined by atomic absorption spectrophotometer (A Analyst 800, PerkinElmer Corporation, Norwalk, Connecticut 06859, USA) followed by the dry ashing method, and the amount of P was determined using an IEO-analyzer (Yellow method).

| | |
|---|---|
| Moisture (%) | 15 |
| pH | 9.42 |
| EC (dsm$^{-1}$) | 4.60 |
| Total C (%) | 22.35 |
| OM (%) | 48.60 |
| C:N ratio | 8.59 |
| Total N (%) | 2.60 |
| Potassium (K. g Kg$^{-1}$) | 12.32 |
| Calcium (Ca, g Kg$^{-1}$) | 12.41 |
| Magnesium (Mg, g Kg$^{-1}$) | 3.40 |
| Sodium (Na, g Kg$^{-1}$) | 2.45 |
| Total P (g Kg$^{-1}$) | 6.30 |
| Copper (mg Kg$^{-1}$) | 41.50 |
| Manganese (Mn, mg Kg$^{-1}$) | 350 |
| Zinc (Zn. Mg Kg$^{-1}$) | 30.72 |

*2.7. Chemical Analysis of Plant Sample*

Plant samples from each treatment were sorted into three categories, shoots (plant components above ground except grains), roots, and grains oven dried for 72 h at 70 °C. A Wiley Hammer Mill with a mesh size of 1 mm was used to crush the oven dried materials in the lab. Total nitrogen (N), phosphate (P), and potassium (K) were measured in the samples. The acid wet digestion method [27] was used to determine the nutrients. In total, 0.25 g of ground materials was transferred to clean 100 mL digestion flasks, and 5 mL of concentrated sulphuric acid ($H_2SO_4$) was added to each flask for the digestion procedure. Next, 2 mL of 50 percent hydrogen peroxide ($H_2O_2$) was added after the samples had been allowed to stand for 2 h. Before cooling, the flasks were heated to 285 °C for 45 min.

This method was repeated twice more to guarantee that all of the food had been digested (colorless). The flasks were then removed from the digesting block, allowed to cool to room temperature, and filled to a capacity of 100 mL with distilled water filtered via filter paper (Whatman no. 1). Before being analyzed for N, P, and K, the digested samples were stored in plastic vials. An Auto Analyzer (AA) (Lachat Instrument, Milwaukee, WI, USA) was used to determine the nitrogen and potassium, while an Automatic Absorption Spectrometer (ASS) was used to determine the potassium, calcium, and magnesium (Perkin Elmer, Massachusetts, 5100, USA).

*2.8. Data Collection*

The data on morphological and yield attributes traits collected in this study include the quantitative qualities that can be measured or quantified using specified measurement procedures. The measured using specific measuring tools such as plant height (PH, cm), panicle length (PL, cm), number of tillers per plant (NT, no.), number of panicles per plant (NP, no.), number of filled grains per panicle (NFG, no.), number of unfilled grains per panicle (UNFG, no), percent filled grain (PFG, %), 1000 grain weight (TGW, g), grain yield (GY, kg ha$^{-1}$), straw yield (SY, kg ha$^{-1}$), biological yield (kg ha$^{-1}$), harvest index (HI,%) photosynthesis rate (PHSYN, $\mu$mol $CO_2$ m$^{-2}$ s$^{-1}$) transpiration rate (TRNSP, $\mu$mol $H_2O$ m$^{-2}$ s$^{-1}$), and grain and straw samples' nutritional content (N, P, and K).

*2.9. Statistical Analysis*

The data were analyzed using a pooled version of Statistical Analysis Software (SAS) version 9.4 to test for significant differences using the analysis of variance (ANOVA) procedure and the least significant differences (LSD) (*p* 0.001, 0.05) to compare the means of the significant characteristics using the Duncan's new multiple range test (DNMRT) [28]. Data were checked for normality and homogeneity of variance before running ANOVA. These were used to determine the level of variance of all observable attributes caused by genotypes, seasons, treatments, genotypes by treatments, genotypes by seasons, and genotypes by treatments by seasons.

**3. Results**

*3.1. Plant Height and Panicle Length*

The significant variation in pant height of rice genotypes at different treatments is shown in Table 4. Application of $T_1$ (100% CFRR) produced the tallest plant height recorded in MR309 (G6) (128.24 cm) followed by Putra2 (G4), Putra1 (G3), and HUA 565 (G1) (126.09 cm, 125.14 cm, and 123.70 cm, respectively) were significantly higher than the other treatments except $T_4$ (10 t ha$^{-1}$ CM), which was similar in plant height. Chicken manure might have increased the soil moisture content, soil porosity, and other plant enhancing characters, and for that reason, increasing the dose of chicken manure increased plant height. Application of $T_2$ (5 tha$^{-1}$ CM) produced shorter plant height recorded in genotype BRRI dhan82 (G9), BRRI dhan39 (G11), BR24 (G7) and BRRI dhan48 (G5) (99.50 cm, 110.05 cm, 110.25 cm, and 110.39 cm, respectively) was presented in Table 5. Increase in plant height in response to the recommended dose of fertilizer might have been primarily due to the improved vegetative growth and supplementary contribution of nitrogen.

In panicle length showed significant differences at the source of variation, which are presented in Table 4. Application of $T_4$ (10 t ha$^{-1}$ CM) produced the longest panicle length in genotype MR309 (G6) (28.89 cm) followed by BRRI dgan72 (G10), MR297 (G2), and Putra1 (G3) (28.87 cm, 27.90 cm, and 27.71 cm, respectively) were significantly longer than the other treatments except $T_3$ (7 t ha$^{-1}$ CM), which produced a similar panicle length. This might be due to a balanced supply of nutrients from chicken manure, which enhanced panicle length. Application of $T_2$ (5 tha$^{-1}$ CM) produced shorter panicle length recorded in BRRI dhan82 (G9), BRRI dhan39 (G11), BRRI dhan75 (G12), and HUA 565(G1) (22.86 cm, 22.92 cm, 23.64 cm, and 23.70 cm, respectively) was presented in Table 5.

### 3.2. Number of Tillers and Panicles

Differences in the number of tillers and number of panicles per meter square showed significant variation ($p \leq 0.01$) among the different treatments and genotypes. Application of $T_4$ (10 t $ha^{-1}$ CM) produced the higher number of tillers and panicles on genotype BRRI dhan75 (G12) (466.64 tillers $m^{-2}$; 422.56 panicles $m^{-2}$) followed by BRRI dhan72 (G10), Putra1 (G3) and MR297 (G2) (446.54 tillers $m^{-2}$, 432.67 tillers $m^{-2}$, 394.18 tillers $m^{-2}$ and 404.27 panicles $m^{-2}$, 394.52 panicles $m^{-2}$, 360.47 panicles $m^{-2}$, respectively) were significantly higher than the other treatments except $T_3$ (7 t $ha^{-1}$ CM) which was recorded similar number of tillers per meter square. Organic sources offer more balanced nutrition to the plants, especially micronutrients which causes better affectivity of tiller in plants grown with chicken manure. Application of $T_2$ (5 $tha^{-1}$ CM) recorded the lowest number of tillers per meter square in BRRI dhan48 (G8), HUA 565 (G1), BR24 (G7), and BRRI dhan39 (G11) (208.39 tillers $m^{-2}$, 246.82 tillers $m^{-2}$, 250.69 tillers $m^{-2}$, 258.66 tillers $m^{-2}$ and 168.96 panicles $m^{-2}$, 204.41 panicles $m^{-2}$, 211.74 panicles $m^{-2}$, and 216.44 panicles $m^{-2}$, respectively) was presented in Table 5.

**Table 4.** A pooled analysis of variance mean square of growth traits across the two seasons.

| Source of Variation | DF | PH | PL | NT | NP | NFG | NUFG | PFG | 1000-GW | GY | SY |
|---|---|---|---|---|---|---|---|---|---|---|---|
| Season (S) | 1 | 337.39 ** | 1.00 ** | 12,880.12 ** | 16,866.72 ** | 2369.01 ** | 2178.06 ** | 256.26 ** | 0.97 ** | 6.13 ** | 0.19 ** |
| Block within S (B/S) | 4 | 28.16 ** | 0.56 ** | 836.35 ** | 575.38 * | 51.08 ns | 9.50 ns | 3.81 ns | 0.02 * | 0.00 ns | 1.03 ** |
| Treatment (T) | 3 | 1032.64 ** | 78.50 ** | 168,592.80 ** | 157,725.29 ** | 12,492.30 ** | 2305.52 ** | 138.32 ** | 3.04 ** | 9.21 ** | 62.03 ** |
| T X S | 3 | 4.78 ns | 0.00 ns | 42.45 ns | 41.50 ns | 96.24 ns | 29.23 ns | 6.55 ns | 0.02 * | 0.03 * | 0.14 ** |
| T X B/S (Error a) | 12 | 1.51 ns | 0.01 ns | 150.75 ns | 339.64 ns | 21.04 ns | 21.08 ns | 4.42 ns | 0.01 * | 0.02 ** | 0.04 ns |
| Genotype (G) | 11 | 642.49 ** | 43.90 ** | 40,910.33 ** | 39,341.22 ** | 4865.34 ** | 1640.28 ** | 231.57 ** | 8.95 ** | 4.95 ** | 3.70 ** |
| G X S | 11 | 2.50 ns | 0.00 ns | 38.73 ns | 30.32 ns | 26.34 ns | 23.69.70 ns | 6.63 * | 0.05 ** | 0.08 ** | 0.13 ** |
| G X T | 33 | 0.91 ns | 0.37 ** | 1921.7 ** | 1986.11 ** | 86.73 ** | 65.98 ** | 13.88 ** | 0.07 ** | 0.35 ** | 1.08 ** |
| G X T X S | 33 | 0.71 ns | 0.00 ns | 56.90 ns | 39.85 ns | 25.29 ns | 3.64 ns | 1.66 ns | 0.04 ** | 0.03 ** | 0.09 ** |
| Error b | 176 | 2.17 | 0.031 | 125.18 | 299.79 | 26.42 | 11.7746 | 3.5 | 0.009 | 0.01 | 0.02 |

| Source of Variation | DF | BY | HI | PHSYN | TRNSP | NG | NS | PG | PS | KG | KS |
|---|---|---|---|---|---|---|---|---|---|---|---|
| Season (S) | 1 | 8.97 ** | 62.93 ** | 0.06 ** | 0.11 ** | 0.004 * | 0.009 ** | 0.002 ** | 0.08 ** | 0.004 ** | 0.007 ** |
| Block within S (B/S) | 4 | 1.064 ** | 9.75 ** | 0.008 ns | 0.02 * | 0.0003 ns | 0.0007 ns | 0.0001 ns | 0.001 ns | 0.0002 ns | 0.0008 ns |
| Treatment (T) | 3 | 115.69 ** | 225.35 ** | 136.81 ** | 14.81 ** | 0.96 ** | 0.72 ** | 0.25 ** | 0.175 ** | 0.21 ** | 0.66 ** |
| T X S | 3 | 0.161 ** | 1.70 ** | 0.17 ns | 0.04 ns | 0.006 ns | 0.07 ns | 0.06 ns | 0.08 ns | 0.05 ns | 0.02 ns |
| T X B/S (Error a) | 12 | 0.07 * | 0.65 * | 0.02 ns | 0.009 ns | 0.0009 ns | 0.0001 ns | 0.0001 ns | 0.0002 ns | 0.0003 ns | 0.0004 ns |
| Genotype (G) | 11 | 15.79 ** | 23.01 ** | 29.73 ** | 3.97 ** | 0.28 ** | 0.25 ** | 0.04 ** | 0.03 ** | 0.029 ** | 0.08 ** |
| G X S | 11 | 0.27 ** | 1.61 ** | 0.52 ns | 0.02 ns | 0.07 ns | 0.06 ns | 0.002 ns | 0.001 ns | 0.05 ns | 0.004 ns |
| G X T | 33 | 1.39 ** | 15.11 ** | 0.40 ** | 0.08 ** | 0.002 ** | 0.004 ** | 0.0003 ** | 0.0006 ** | 0.001 ** | 0.0007 ** |
| G X T X S | 33 | 0.13 ** | 1.08 ** | 0.02 ns | 0.005 ns | 0.06 ns | 0.02 ns | 0.008 ns | 0.003 ns | 0.05 ns | 0.004 ns |
| Error b | 176 | 0.03 | 0.29 | 0.0150 | 0.0070 | 0.0004 | 0.0003 | 0.0002 | 0.0002 | 0.0002 | 0.0003 |

Legend: ** highly significant at $p \leq 0.01$ level, * significant at $p \leq 0.05$ level, ns = non-significant, DF = Degree of freedom, PH = Plant height, PL = Panicle length, NT = Number of tiller, NP = Number of panicle, NFG = Number of filled grain, NUFG = Number of unfilled grain, PFG = Percent filled grain, 1000-GW = 1000-grain weight, GY = grain yield, SY = straw yield, BY = Biological yield, HI = Harvest index, PHSYN = Photosynthesis rate, TRNSP = Transpiration rate, NG = Nitrogen content of grain, NS = Nitrogen content of straw, PG = Phosphorus content of grain, PS = Phosphorus content of straw, KG = Potassium content of grain, and KS = Potassium content of straw.

**Table 5.** Plant height (cm) and Panicle length (cm) of 12 rice genotypes as influenced by treatments (pooled over two seasons).

| Genotype (G) | Treatment (T) | PH | PL | NT | NP | NFG | NUFG |
|---|---|---|---|---|---|---|---|
| G1 | 1 | 123.7 a | 24.76 b | 277.48 c | 242.69 c | 159.43 a | 40.72 b |
| | 2 | 114.62c | 23.70 c | 246.82 d | 211.74 d | 158.75 a | 39.64 b |
| | 3 | 117.85 b | 25.54 a | 324.56 b | 286.338 b | 153.26 a | 51.89 a |
| | 4 | 121.04 a | 25.88 a | 377.78 a | 338.43 a | 175.38 a | 50.38 a |
| | LSD (0.05) | 2.82 | 0.37 | 22.74 | 27.82 | 30.28 | 6.14 |
| G2 | 1 | 119.93 a | 26.71 b | 358.58 bc | 317.37 c | 180.26 b | 36.47 b |
| | 2 | 110.55 b | 26.17 c | 343.22 c | 300.63 d | 164.78 c | 30.87 c |
| | 3 | 113.64 b | 27.68 a | 372.79 b | 336.82 b | 181.74 b | 45.72 a |
| | 4 | 117.26 a | 27.90 a | 394.18 a | 360.47 a | 193.22 a | 43.58 a |
| | LSD (0.05) | 3.31 | 0.43 | 17.82 | 11.8 | 7.35 | 6.24 |
| G3 | 1 | 125.14 a | 24.98 c | 368.62 c | 328.44 c | 187.37 b | 50.36 b |
| | 2 | 116.37 c | 24.22 c | 354.78 c | 310.55 d | 166.24 c | 42.74 b |
| | 3 | 119.77 b | 27.42 a | 400.35 b | 358.14 b | 192.35 b | 55.83 a |
| | 4 | 123.47 a | 27.71 a | 432.67 a | 394.52 a | 207.46 a | 59.45 a |
| | LSD (0.05) | 2.35 | 0.66 | 14.86 | 8.22 | 7.29 | 7.76 |
| G4 | 1 | 126.09 a | 26.62 b | 288.45 b | 248.79 ab | 195.27 c | 50.76 ab |
| | 2 | 116.21 c | 24.88 c | 262.37 b | 220.37 b | 173.66 d | 48.61 b |
| | 3 | 119.65 b | 26.78 a | 291.36 b | 246.39 ab | 206.35 b | 53.85 ab |
| | 4 | 124.67 a | 27.23 a | 393.48 a | 323.06 a | 214.6 a | 54.38 a |
| | LSD (0.05) | 2.84 | 0.47 | 51.49 | 77.57 | 6.93 | 6.07 |
| G5 | 1 | 122.34 c | 25.43 c | 299.34 c | 262.54 ab | 184.52 c | 40.37 b |
| | 2 | 112.78 c | 24.75 d | 268.48 d | 228.42 b | 165.28 d | 41.69 b |
| | 3 | 116.96 b | 26.77 b | 336.62 b | 302.69 a | 194.47 b | 44.78 ab |
| | 4 | 121.59 a | 26.95 a | 382.58 a | 307.88 a | 208.14 a | 49.36 a |
| | LSD (0.05) | 1.88 | 0.17 | 17.15 | 51.77 | 6.72 | 5.80 |
| G6 | 1 | 128.24 a | 27.58 c | 290.49 c | 248.46 c | 177.43 b | 32.48 b |
| | 2 | 119.94 c | 26.89 d | 269.35 d | 226.37 d | 161.82 c | 25.28 c |
| | 3 | 122.93 b | 28.70 b | 340.64 b | 303.72 b | 182.25 b | 40.46 a |
| | 4 | 124.06 a | 28.89 a | 384.24 a | 352.48 a | 193.74 a | 42.54 a |
| | LSD (0.05) | 2.49 | 0.12 | 14.86 | 7.53 | 6.93 | 6.45 |
| G7 | 1 | 118.86 a | 24.58 c | 275.66 c | 240.86c | 164.58 c | 26.49 c |
| | 2 | 110.25 c | 23.82 d | 250.69 d | 204.41 d | 142.79d | 27.88 bc |
| | 3 | 113.3 bc | 25.67 b | 286.49 b | 307.52 b | 172.89 b | 32.77 ab |
| | 4 | 116.28 ab | 25.96 a | 314.57 a | 341.08 a | 179.36 a | 37.25 a |
| | LSD (0.05) | 4.27 | 0.22 | 15.06 | 13.86 | 4.98 | 5.72 |
| G8 | 1 | 119.14 a | 24.66 c | 272.86 c | 232.72 a | 179.45 b | 29.54 b |
| | 2 | 110.39 c | 23.88 d | 208.39 d | 168.96 b | 165.38 c | 29.84 b |
| | 3 | 112.94 bc | 25.74 b | 286.34 b | 236.49 a | 184.74 b | 32.68 b |
| | 4 | 116.49 ab | 25.92 a | 308.76 a | 267.38 a | 197.69 a | 43.75 a |
| | LSD (0.05) | 3.94 | 0.13 | 61.04 | 56.83 | 6.54 | 4.92 |
| G9 | 1 | 109.86 a | 23.78 c | 290.38 c | 252.47 c | 149.27 c | 30.48 bc |
| | 2 | 99.50 d | 22.86 d | 269.84 d | 235.64 c | 134.86 d | 29.66 c |
| | 3 | 103.24 c | 24.77 b | 318.68 b | 283.44 b | 160.76 b | 37.49 ab |
| | 4 | 106.98 b | 25.18 a | 365.14 a | 324.16 a | 169.28 a | 36.96 a |
| | LSD (0.05) | 2.66 | 0.26 | 11.43 | 18.69 | 5.15 | 5.84 |
| G10 | 1 | 122.32 a | 27.56 c | 353.74 c | 309.74 c | 168.33 b | 42.78 bc |
| | 2 | 114.33 c | 26.78 d | 272.43 d | 230.77 d | 157.53 c | 36.89 c |
| | 3 | 116.77 bc | 28.57 b | 400.37 b | 364.58 b | 178.46 a | 49.54 ab |
| | 4 | 119.51 ab | 28.87 a | 446.54 a | 404.27 a | 183.65 a | 51.49 a |
| | LSD (0.05) | 3.02 | 0.21 | 16.52 | 16.08 | 7.16 | 6.78 |
| G11 | 1 | 119.83a | 23.72 c | 279.65 b | 250.87 b | 170.34 c | 27.75 b |
| | 2 | 110.05 b | 22.92 d | 258.66 c | 216.44 c | 162.42 d | 29.85 b |
| | 3 | 112.54 b | 24.79 b | 299.49 b | 258.78 b | 184.29 b | 44.72 a |
| | 4 | 116.85 a | 25.36 a | 332.76 a | 294.36 a | 189.11 a | 42.16 a |
| | LSD (0.05) | 4.06 | 0.34 | 20.27 | 15.75 | 6.24 | 6.54 |
| G12 | 1 | 122.58 a | 24.15 b | 360.92 c | 311.46 c | 192.43 b | 32.55 b |
| | 2 | 113.07 c | 23.64 b | 332.28 d | 296.69 c | 178.64 c | 25.89 a |
| | 3 | 116.38 bc | 25.12 a | 430.45 b | 396.28 b | 216.55 a | 32.48 a |
| | 4 | 119.79 ab | 25.6 a | 466.64 a | 422.56 a | 224.49 a | 31.79 a |
| | LSD (0.05) | 4.12 | 0.59 | 18.53 | 19.62 | 8.62 | 5.12 |

Means in a column followed by the same latter are not significantly different at *p* = 0.05 using Duncan's new multiple range test (DNMRT); $T_1$ = 100% chemical fertilizer recommended rate (CFRR), $T_2$ = 5 t/ha chicken manure (CM), $T_3$ = 7 t/ha CM, and $T_4$ = 10 t/ha CM; PH = Plant height; PL = Panicle length; NT = Number of tillers; NP = Number of panicles; NFG = Number of filled grains; and NUFG = Number of unfilled grains.

### 3.3. Number of Filled Grains and Unfilled Grains

Different doses of chicken manure had shown significant variations ($p \leq 0.01$) on filled and unfilled grains per panicle of the rice genotypes, as shown in Table 4. Filled and unfilled grains panicle-$^1$ of the rice genotypes were significantly influenced by different treatments. Application of $T_4$ (10 t ha$^{-1}$ CM) produced the highest number of filled grains in genotype BRRI dhan75 (G12) (224.49 filled grain panicle$^{-1}$) followed by Putra2 (G4), MR309 (G5), and Putra1 (G3) (214.60 filled grain panicle$^{-1}$, 208.14 filled grains panicle$^{-1}$, 207.46 filled grain panicle$^{-1}$) and higher number of unfilled grains in genotype Putra1 (G3) (59.45 unfilled grains panicle$^{-1}$) followed by Putra2 (G4), BRRI dhan72 (G10), and HUA 565 (G1) (54.38 unfilled grains panicle$^{-1}$, 51.49 unfilled grains panicle$^{-1}$, 50.38 unfilled grains panicle$^{-1}$, respectively) were significantly higher than other treatments except $T_3$ (7 t ha$^{-1}$ CM), for which similar numbers of filled and unfilled grains per panicle were recorded. Application of $T_2$ (5 tha$^{-1}$ CM) recorded the lowest number of filled grains per panicle in BRRI dhan82 (G9), BR24 (G7), BRRI dhan72 (G10), and HUA 565 (G1) (134.86 filled grain panicle$^{-1}$, 142.79 filled grains panicle$^{-1}$, 157.53 filled grains panicle$^{-1}$, and 158.75 filled grains panicle$^{-1}$) and the lowest number of unfilled grains per panicle in MR309 (G6), BRRI dhan75 (G12), BR24 (G7), and BRRI dhan2 (G9) (25.28 unfilled grains panicle$^{-1}$, 25.89 unfilled grain panicle$^{-1}$, 27.88 unfilled grains panicle$^{-1}$, and 29.66 unfilled grains panicle$^{-1}$, respectively) was presented in Table 5.

### 3.4. Percent Filled Grain and 1000-Grain Weight

Different levels of chicken manure had shown significant variations ($p \leq 0.01$) on the percent filled grains and 1000-grain weight of the rice genotypes, as shown in Table 4. Percent filled grains and 1000-grain weight of the rice genotypes were significantly influenced by different level of treatments. 1000-grain weight is mostly mediated by genetic potential, but in this case, it declines significantly with decreased application of chicken manure, as well as with inorganic treatment due to a severe deficiency of essential nutrients, because of which the plants failed to produce a bold grain. Results indicated that application of $T_4$ (10 t ha$^{-1}$ CM) produced the higher percent filled grains in genotype BRRI dhan75 (G12) (87.83%) followed by BRRI dhan39 (G11), BR24 (G7), and MR309 (G6) (87.72%, 85.80%, and 85.64%) and higher 1000-grain weight in genotype BRRI dhan39 (G11) (27.76 g) followed by Putra2 (G3), BRRI dhan72 (G10) and MR303 (G5) (27.70 g, 27.55 g, and 27.44 g, respectively) were significantly higher than the other treatments except $T_3$ (7 t ha$^{-1}$ CM), for which similar percent filled grains and 1000-grain weight. Application of $T_2$ (5 tha$^{-1}$ CM) recorded the lowest percent filled grains in Putra2 (G4), Putra1 (G46), BRRI dhan72 (G10), and MR303 (G5) (72.32%, 73.49%, 76.22%, and 77.83%) and the lowest 1000-grain weight in BRRI dhan48 (G8), HUA 565 (G1), BRRI dhan82 (G9) and BRRI dhan39 (G11) (25.33 g, 25.43 g, 25.75 g, and 26.33 g, respectively) was presented in Table 6.

### 3.5. Grain Yield and Straw Yield

Different doses of chicken manure showed significant variation ($p \leq 0.01$) in grain yield and straw yield of rice genotypes, as shown in Table 4. Grain yield and straw yield of rice genotypes were significantly influenced by different doses of chicken manure. The yield advantages due to the integration of organic sources and inorganic fertilizers over chemical fertilizers alone might be due to the availability of nutrients for a shorter period, as the mineralization of nitrogen is more rapid and in turn the losses of inorganic nitrogen due to volatilization, denitrification, leaching, etc. Chicken manure may increase the vegetative growth of plants and thereby increase the straw yield of rice. The results showed that application of $T_4$ (10 t ha$^{-1}$ CM) produced the higher grain yield in genotype BRRI dhan75 (G12) (8.02 t ha$^{-1}$) followed by BRRI dhan72 (G10), HUA 565 (G1), and Putra2 (G4) (7.81 t ha$^{-1}$, 7.76 t ha$^{-1}$, and 7.72 t ha$^{-1}$) and the higher straw yield in genotype BRRI dhan39 (G11) (9.88 t ha$^{-1}$) followed by MR309 (G6), Putra1 (G3) and MR303 (G5) (9.86 t ha$^{-1}$, 9.35 t ha$^{-1}$, and 9.30 t ha$^{-1}$, respectively) were significantly higher than the other treatments except $T_3$ (7 t ha$^{-1}$ CM), which recorded similar grain and straw yield per

hectare. Application of $T_2$ (5 tha$^{-1}$ CM) recorded the lowest grain yields in BRRI dhan82 (G9), MR303 (G5), BR24 (G7), and BRRI dhan39 (G11) (5.66 t ha$^{-1}$, 5.97 t ha$^{-1}$, 5.98 t ha$^{-1}$, and 6.47 t ha$^{-1}$) and the lowest straw yields in BRRI dhan82 (G9), BR24 (G7), MR303 (G5), and MR297 (G2) (5.21 t ha$^{-1}$, 5.36 t ha$^{-1}$, 5.77 t ha$^{-1}$ and 6.19 t ha$^{-1}$, respectively) (Table 6).

### 3.6. Biological Yield and Harvest Index

Different doses of chicken manure is significant variations ($p \leq 0.01$) in the biological yield per hectare and harvest index of rice genotypes, as shown in Table 4. Biological yield and harvest index were significantly influenced by different doses of chicken manure. Higher biological yield might be due to the increase in growth and yield attributes. Organic matter provided micronutrients and increased the cation exchange capacity of the soil, and thus improved nutrient availability. Higher yield and harvest index due to chicken manure indicates a better portioning of photosynthetic substance to economic yield. The appreciably high harvest index shows the efficiency of converting biological yield into economic yield. Application of $T_4$ (10 t ha$^{-1}$ CM) produced the highest biological yield in genotype MR309 (G6) (17.36 t ha$^{-1}$) followed by Putra2 (G4), BRRI dhan75 (G12), and Putra1 (G3) (17.11 t ha$^{-1}$, 16.95 t ha$^{-1}$, and 16.75 t ha$^{-1}$) and the highest harvest index in genotype MR303 (G5) (50.64%) followed by BRRI dhan39 (G11), Putra1 (G3), and BRRI dhan72 (G10) (49.97%, 49.55%, and 48.91%, respectively) were significantly higher than the other treatments except $T_3$ (7 t ha$^{-1}$ CM), for which was recorded a similar biological yield and harvest index. Application of $T_2$ (5 tha$^{-1}$ CM) recorded the lowest biological yields in BRRI dhan82 (G9), BR24 (G7), BRRI dhan39 (G11), and MR303 (G5) (11.97 t ha$^{-1}$, 12.78 t ha$^{-1}$, 13.10 t ha$^{-1}$, and 13.24 t ha$^{-1}$) and the lowest harvest indexes in MR309 (G6), MR297 (G2), BR24 (G7), and BRRI dhan48 (G8) (40.80%, 41.02%, 41.33%, and 41.49%, respectively) (Table 6).

### 3.7. Photosynthesis and Transpiration Rate

The photosynthesis and transpiration rate showed significant differences ($p \leq 0.01$) among the rice treatment, genotype, season, and genotype by treatment, as presented in Table 4. The treatment by season, genotype by season, and genotype by treatment by season were not significant. The results showed that application of $T_4$ (10 t ha$^{-1}$ CM) produced the highest photosynthesis rate in genotype MR303 (G5) (23.69 μmol $CO_2$ m$^{-2}$ s$^{-1}$) followed by BR24 (G7), MR297 (G2), and BRRI dhan72 (G10) (23.48 μmol $CO_2$ m$^{-2}$ s$^{-1}$, 23.03 μmol $CO_2$ m$^{-2}$ s$^{-1}$, and 23.01 μmol $CO_2$ m$^{-2}$ s$^{-1}$) and the highest transpiration rate in genotype BRRI dhan72 (G10) (7.75 mmol $H_2O$ m$^{-2}$ s$^{-1}$) followed by MR297 (G2), Putra2 (G4), and BR24 (G7) (7.16 mmol $H_2O$ m$^{-2}$ s$^{-1}$, 7.02 mmol $H_2O$ m$^{-2}$ s$^{-1}$, and 7.00 mmol $H_2O$ m$^{-2}$ s$^{-1}$), which were significantly higher than the other treatments except $T_3$ (7 t ha$^{-1}$ CM), for which similar photosynthesis and transpiration rates were recorded. Application of $T_1$ (100% CFRR) produced the lowest photosynthesis rate in BRRI dhan82 (G9), BRRI dhan39 (G11), BRRI dhan48 (G8), and MR309 (G6) (13.56 μmol $CO_2$ m$^{-2}$ s$^{-1}$, 15.64 μmol $CO_2$ m$^{-2}$ s$^{-1}$, 15.70 μmol $CO_2$ m$^{-2}$ s$^{-1}$, and 16.30 μmol $CO_2$ m$^{-2}$ s$^{-1}$) and the lowest transpiration rate in BRRI dhan82 (G9), BRRI dhan48 (G8), BRRI dhan39 (G11), and BR24 (G7) (3.98 mmol $H_2O$ m$^{-2}$ s$^{-1}$, 4.66 mmol $H_2O$ m$^{-2}$ s$^{-1}$, 4.92 mmol $H_2O$ m$^{-2}$ s$^{-1}$, and 4.96 mmol $H_2O$ m$^{-2}$ s$^{-1}$ respectively), was presented in Table 6.

**Table 6.** Percent filled grain (%) and 1000-grain weight (g) of 12 rice genotypes as influenced by treatments (pooled over two seasons).

| Genotype (G) | Treatment (T) | PFG | 1000-GW | GY | SY | BY | HI | PHSYN | TRNSP |
|---|---|---|---|---|---|---|---|---|---|
| G1 | 1 | 80.84 b | 25.78 ab | 6.97 c | 7.97 b | 14.95 c | 45.65 b | 17.53 d | 5.02 c |
| | 2 | 78.06 c | 25.43 b | 6.83 c | 7.72 c | 14.55 b | 42.96 c | 19.84 c | 5.71 b |
| | 3 | 82.66 a | 25.92 ab | 7.23 b | 7.94 b | 15.18 b | 47.66 a | 20.17 b | 5.69 b |
| | 4 | 83.78 a | 26.25 a | 7.76 a | 8.92 a | 16.68 a | 48.5 a | 22.54 a | 6.47 a |
| LSD (0.05) | | 11.60 | 0.06 | 0.18 | 0.21 | 0.27 | 0.84 | 0.19 | 0.14 |
| G2 | 1 | 81.26 b | 26.81 a | 6.67 c | 7.56 c | 14.22 c | 44.785 b | 18.65 d | 5.79 c |
| | 2 | 79.53 a | 26.53 b | 6.56 d | 7.18 d | 13.53 d | 41.02 c | 20.76 c | 6.44 d |
| | 3 | 83.05 c | 26.89 a | 6.94 b | 8.14 b | 15.09 b | 46.04 a | 21.17 b | 6.94 b |
| | 4 | 84.68 a | 26.92 a | 7.33 a | 8.94 a | 16.28 a | 47.11 a | 23.03 a | 7.16 a |
| LSD (0.05) | | 2.05 | 0.07 | 0.15 | 0.22 | 0.21 | 1.01 | 0.18 | 0.16 |
| G3 | 1 | 75.75 b | 27.33 b | 6.76 b | 7.66 c | 14.42 c | 46.9 c | 16.38 d | 5.56 c |
| | 2 | 73.49 c | 27.26 c | 6.7 b | 6.79 d | 13.49 d | 43.68 d | 18.97 c | 6.14 d |
| | 3 | 78.32 a | 27.66 a | 7.42 a | 8.97 b | 16.47 a | 47.16 b | 19.22 b | 6.37 b |
| | 4 | 80.92 a | 27.70 a | 7.49 a | 9.35 a | 16.75 b | 49.55 a | 21.63 a | 6.78 a |
| LSD (0.05) | | 2.35 | 0.06 | 0.21 | 0.13 | 0.24 | 0.77 | 0.17 | 0.14 |
| G4 | 1 | 75.56 b | 27.11 c | 6.87 c | 7.33 c | 14.20 c | 45.37 b | 18.79 c | 5.96 c |
| | 2 | 72.32 c | 26.87 d | 6.85 c | 7.29 c | 14.15 c | 43.43 c | 21.04 b | 6.23 d |
| | 3 | 79.45 a | 27.34 b | 7.29 b | 8.69 b | 16.39 b | 47.47 a | 21.25 b | 6.87 b |
| | 4 | 79.77 a | 27.39 a | 7.72 a | 9.88 a | 17.11 a | 48.12 a | 22.56 a | 7.02 a |
| LSD (0.05) | | 2.32 | 0.05 | 0.14 | 0.17 | 0.27 | 0.75 | 0.18 | 0.11 |
| G5 | 1 | 80.96 b | 27.08 b | 6.29 b | 7.26 c | 13.82 d | 45.11 c | 18.73 d | 4.98 c |
| | 2 | 77.83 c | 26.81 c | 5.97 c | 6.86 c | 13.24 c | 42.4 d | 21.76 c | 5.77 d |
| | 3 | 81.35 a | 27.38 a | 6.57 b | 8.15 b | 14.45 b | 47.59 b | 22.11 b | 6.08 b |
| | 4 | 82.89 a | 27.44 a | 6.73 a | 9.3 a | 15.67 a | 50.64 a | 23.69 a | 6.74 a |
| LSD (0.05) | | 2.04 | 0.07 | 0.19 | 0.48 | 0.49 | 1.69 | 0.21 | 0.18 |
| G6 | 1 | 82.69 bc | 26.96 a | 6.65 c | 7.66 c | 14.31 c | 43.48 c | 17.59 d | 5.46 c |
| | 2 | 79.44 d | 26.7 b | 6.52 c | 7.42 d | 13.95 d | 40.8 d | 20.03 c | 5.98 d |
| | 3 | 83.79 b | 26.92 a | 7.31 b | 8.62 b | 16.20 b | 45.58 b | 20.25 b | 6.3 b |
| | 4 | 85.64 a | 26.95 a | 7.56 a | 9.86 a | 17.36 a | 46.74 a | 21.97 a | 6.88 a |
| LSD (0.05) | | 2.34 | 0.06 | 0.21 | 0.15 | 0.26 | 0.85 | 0.19 | 0.13 |
| G7 | 1 | 82.34 b | 26.62 c | 6.15 c | 7.84 c | 13.80 c | 44.45 ab | 17.74 d | 4.96 c |
| | 2 | 80.92 c | 26.67 d | 5.98 d | 6.77 d | 12.78 d | 41.33 c | 20.85 c | 5.84 d |
| | 3 | 85.11 a | 26.77 b | 6.62 b | 8.14 b | 14.73 b | 46.89 a | 21.43 b | 6.27 b |
| | 4 | 85.80 a | 26.84 a | 7.03 a | 9.25 a | 16.28 a | 47.18 a | 23.48 a | 7.00 a |
| LSD (0.05) | | 2.57 | 0.05 | 0.25 | 0.14 | 0.31 | 0.96 | 0.15 | 0.11 |
| G8 | 1 | 82.83 c | 25.68 b | 6.36 b | 7.06 b | 13.42 c | 44.39 b | 15.70 d | 4.66 d |
| | 2 | 79.95 d | 25.33 b | 6.48 b | 7.16 b | 13.64 c | 41.49 c | 18.01 c | 4.69 c |
| | 3 | 83.4 b | 25.63 b | 6.55 b | 9.01 a | 15.57 b | 47.09 a | 18.26 b | 5.28 b |
| | 4 | 84.06 a | 26.37 a | 7.1 a | 9.14 a | 16.22 a | 48.82 a | 20.61 a | 5.96 a |
| LSD (0.05) | | 1.88 | 0.05 | 0.19 | 0.21 | 0.22 | 1.12 | 0.23 | 0.12 |
| G9 | 1 | 80.17 c | 25.84 b | 5.75 c | 6.97 c | 12.64 c | 43.83 b | 13.56 d | 3.98 c |
| | 2 | 77.88 d | 25.75 c | 5.66 c | 6.23 d | 11.97 d | 41.99 c | 16.64 c | 5.09 d |
| | 3 | 82.95 b | 25.84 b | 6.29 b | 7.34 b | 13.54 b | 45.76 ab | 17.35 b | 5.42 b |
| | 4 | 83.44 a | 26.05 a | 6.48 a | 7.79 a | 14.26 a | 47.25 a | 19.22 a | 5.88 a |
| LSD (0.05) | | 2.57 | 0.06 | 0.18 | 0.11 | 0.30 | 0.47 | 0.17 | 0.12 |
| G10 | 1 | 79.77 b | 27.31 b | 7.12 c | 7.68 c | 14.80 b | 45.11 b | 18.74 d | 6.05 b |
| | 2 | 76.22 c | 26.86 c | 6.79 d | 6.70 d | 13.49 b | 42.36 c | 21.70 c | 6.44 b |
| | 3 | 80.31 a | 27.54 a | 7.42 b | 9.04 a | 16.47 a | 47.08 a | 21.93 b | 6.44 b |
| | 4 | 81.26 a | 27.55 a | 7.81 a | 8.94 b | 16.65 a | 48.91 a | 23.01 a | 7.75 a |
| LSD (0.05) | | 2.11 | 0.07 | 0.20 | 0.16 | 0.25 | 0.78 | 0.17 | 0.13 |
| G11 | 1 | 83.35 c | 26.53 c | 6.85 c | 7.51 c | 14.36 c | 45.68 c | 15.64 d | 4.92 c |
| | 2 | 79.32 d | 26.33 d | 6.47 b | 6.65 d | 13.10 d | 43.23 d | 17.72 c | 5.22 d |
| | 3 | 85.5 cb | 27.59 a | 7.05 b | 8.76 b | 15.82 b | 47.58 b | 18.34 b | 5.74 b |
| | 4 | 87.72 a | 27.76 b | 7.44 a | 9.18 a | 16.62 a | 49.97a | 20.56 a | 6.18 a |
| LSD (0.05) | | 3.06 | 0.06 | 0.16 | 0.2 | 0.24 | 1.01 | 0.26 | 0.12 |
| G12 | 1 | 84.69 b | 26.78 ab | 7.14 b | 8.27 b | 15.41 b | 44.31 bc | 16.3 d | 5.11 c |
| | 2 | 80.55 c | 26.50 c | 6.69 c | 7.55 c | 14.25 c | 43.98 c | 17.62 c | 5.58 d |
| | 3 | 87.1 a | 26.83 b | 7.24 b | 8.93 a | 16.70 a | 46.72 b | 18.46 b | 5.94 b |
| | 4 | 87.83 a | 26.90 a | 8.02 a | 9.12 a | 16.95 a | 47.32 a | 19.97 a | 6.28 a |
| LSD (0.05) | | 2.07 | 0.06 | 0.23 | 0.49 | 0.35 | 1.97 | 0.18 | 0.15 |

Means in a column followed by the same latter are not significantly different at *p* = 0.05 using Duncan's new multiple range test (DNMRT); $T_1$ = 100% chemical fertilizer recommended rate (CFRR); $T_2$ = 5 t/ha chicken manure (CM), $T_3$ = 7 t/ha CM, and $T_4$ = 10 t/ha CM; PFG = Percent filled grain; 1000-GW = 1000-grain weight; GY = grain yield; SY = straw yield, BY = Biological yield, HI = Harvest index, PHSYN = Photosynthesis rate, and TRNSP = Transpiration rate.

### 3.8. N Content in Grains and Straw

The nitrogen content in grain and straw showed significant differences ($p \leq 0.01$) among the rice treatment, genotype, season, and genotype by treatment as presented in Table 4. The treatment by season, genotype by season and genotype by treatment by season were not significant. Results showed that application of $T_4$ (10 t ha$^{-1}$ CM) produced the highest N content in grains in genotype BRRI dhan72 (G10) (1.86%) followed by Putra2 (G4), Putra1 (G3), and MR297 (G2) (1.84%, 1.82%, and 1.81%) and the highest N content in straw in genotype BRRI dhan82 (G9) (1.76%) followed by BRRI dhan72 (G10), Putra2 (G4), and MR297 (G2) (1.75%, 1.67%, and 1.66%, respectively) were significantly higher than the other treatments except $T_3$ (7 t ha$^{-1}$ CM), for which similar N content in grains and straw was recorded. Application of $T_1$ (100% CFRR) produced the lowest N content in grains in BRRI dhan82 (G9), HUA 565 (G1), BRRI dhan75 (G12), and MR303 (G5) (1.09%, 1.18%, 1.25%, and 1.28%) and the lowest N content in straw in BRRI dhan82 (G9), HUA 565 (G1), BRRI dhan48 (G8), and BRRI dhan75 (G12) (1.01%, 1.03%, 1.08%, and 1.11%, respectively), as presented in Table 7.

### 3.9. P Content in Grains and Straw

The phosphorus content in grain and straw showed significant differences ($p \leq 0.01$) related to the rice treatment, genotype, season, and genotype by treatment as presented in Table 4. The treatment by season, genotype by season, and genotype by treatment by season were not significant. Results showed that application of $T_4$ (10 t ha$^{-1}$ CM) produced the highest P content in grains in genotype MR297 (G2) (0.49%) followed by BRRI dhan72 (G10), Putra1 (G3), and MR303 (G5) (0.46%, 0.44%, and 0.43%) and the highest P content in straw in genotype BRRI dhan72 (G10) (0.44%) followed by BR24 (G7), MR296 (G2), and Putra1 (G3) (0.43%, 0.42%, and 0.38%, respectively) were significantly higher than the other treatments except $T_3$ (7 t ha$^{-1}$ CM), for which a similar P content in grains and straw was recorded. Application of $T_1$ (100% CFRR) produced the lowest P content in grains in BRRI dhan82 (G9), BRRI dhan75 (G12), MR309 (G6), and HUA 565 (G1) (0.14%, 0.15%, 0.17%, and 0.18%) and the lowest P content in straw in BRRI dhan82 (G9), MR309 (G6), HUA 565 (G1), and BRRI dhan48 (G8) (0.11%, 0.12%, 0.12%, and 0.16%, respectively) was presented in Table 7.

### 3.10. K Content in Grains and Straw

The potassium content in grain and straw showed significant differences ($p \leq 0.01$) related to the rice treatment, genotype, season, genotype by treatment as presented in Table 4. The treatment by season, genotype by season, and genotype by treatment by season were not significant. Results showed that application of $T_4$ (10 t ha$^{-1}$ CM) produced the highest K content in grains in genotype MR297 (G2) (0.57%) followed by BRRI dhan39 (G11), BRRI dhan72 (G10), and MR303 (G8) (0.57%, 0.56%, and 0.54%) and the highest K content in straw in genotype Putra2 (G4) (2.18%) followed by BRRI dhan72 (G10), MR303 (G5), and BRRI dhan75 (G12) (2.18%, 2.16%, and 2.15%, respectively) were significantly higher than the other treatments except $T_3$ (7 t ha$^{-1}$ CM), for which a similar K content in grains and straw was recorded. Application of $T_1$ (100% CFRR) produced the lowest K content in grains in BRRI dhan82 (G9), HUA 565 (G1), MR309 (G6), and Putra2 (G4) (0.22%, 0.25%, 0.26%, and 0.26%) and the lowest K content in straw in BRRI dhan82 (G9), HUA 565 (G1), BRRI dhan75 (G12), and BRRI dhan39 (G11) (1.57%, 1.67%, 1.74%, and 1.76%, respectively) was presented in Table 7.

**Table 7.** Nitrogen content in grain and straw (%) of 12 rice genotypes as influenced by genotype under organic fertilizer treatments for 1st and 2nd planting season.

| Genotype (G) | Treatment (T) | NG | NS | PG | PS | KG | KS |
|---|---|---|---|---|---|---|---|
| G1 | 1 | 1.18 d | 1.03 c | 0.18 d | 0.12 d | 0.25 d | 1.67 d |
| | 2 | 1.22 c | 1.19 d | 0.21 c | 0.17 c | 0.31 c | 1.75 c |
| | 3 | 1.45 b | 1.27 b | 0.26 b | 0.21 b | 0.38 b | 1.84 b |
| | 4 | 1.53 a | 1.34 a | 0.35 a | 0.32 a | 0.44 a | 2.01 a |
| | LSD (0.05) | 0.03 | 0.03 | 0.02 | 0.03 | 0.02 | 0.03 |
| G2 | 1 | 1.50 d | 1.36 d | 0.27 d | 0.21 d | 0.38 d | 1.76 d |
| | 2 | 1.57 c | 1.47 c | 0.35 c | 0.33 c | 0.43 c | 1.83 c |
| | 3 | 1.74 b | 1.60 b | 0.44 b | 0.38 b | 0.48 b | 1.93 b |
| | 4 | 1.81 a | 1.66 a | 0.49 a | 0.42 a | 0.57 a | 2.05 a |
| | LSD (0.05) | 0.02 | 0.03 | 0.03 | 0.02 | 0.02 | 0.04 |
| G3 | 1 | 1.45 d | 1.29 d | 0.25 c | 0.21 d | 0.36 c | 1.84 c |
| | 2 | 1.51 c | 1.36 c | 0.27 c | 0.25 c | 0.38 c | 1.90 b |
| | 3 | 1.70 b | 1.54 b | 0.36 b | 0.31 b | 0.44 b | 2.02 a |
| | 4 | 1.82 a | 1.64 a | 0.44 a | 0.38 a | 0.52 a | 2.05 a |
| | LSD (0.05) | 0.03 | 0.02 | 0.02 | 0.03 | 0.03 | 0.03 |
| G4 | 1 | 1.46 d | 1.26 d | 0.19 d | 0.17 c | 0.26 d | 1.88 d |
| | 2 | 1.54 c | 1.42 c | 0.26 c | 0.22 bc | 0.37 c | 1.95 c |
| | 3 | 1.71 b | 1.64 b | 0.29 b | 0.26 b | 0.41 b | 2.06 b |
| | 4 | 1.84 a | 1.67 a | 0.37 a | 0.32 a | 0.52 a | 2.18 a |
| | LSD (0.05) | 0.02 | 0.02 | 0.03 | 0.03 | 0.02 | 0.02 |
| G5 | 1 | 1.28 d | 1.17 d | 0.24 d | 0.23 b | 0.30 d | 1.86 d |
| | 2 | 1.36 c | 1.27 c | 0.28 c | 0.23 b | 0.34 c | 1.94 c |
| | 3 | 1.47 b | 1.32 b | 0.35 b | 0.31 a | 0.43 b | 2.02 b |
| | 4 | 1.58 a | 1.53 a | 0.43 a | 0.35 a | 0.54 a | 2.16 a |
| | LSD (0.05) | 0.03 | 0.03 | 0.02 | 0.03 | 0.01 | 0.02 |
| G6 | 1 | 1.37 d | 1.15 d | 0.17 d | 0.12 d | 0.26 c | 1.84 d |
| | 2 | 1.41 c | 1.32 c | 0.22 c | 0.17 c | 0.37 b | 1.94 c |
| | 3 | 1.54 b | 1.4 b | 0.28 b | 0.21 b | 0.39 b | 2.05 b |
| | 4 | 1.71 a | 1.55 a | 0.37 a | 0.3 a | 0.44 a | 2.14 a |
| | LSD (0.05) | 0.03 | 0.03 | 0.02 | 0.03 | 0.02 | 0.03 |
| G7 | 1 | 1.47 d | 1.36 d | 0.22 d | 0.18 d | 0.33 d | 1.78 d |
| | 2 | 1.55 c | 1.45 c | 0.29 c | 0.27 c | 0.38 c | 1.87 c |
| | 3 | 1.62 b | 1.49 b | 0.37 b | 0.32 b | 0.45 b | 2.01 b |
| | 4 | 1.74 a | 1.58 a | 0.42 a | 0.43 a | 0.51 a | 2.08 a |
| | LSD (0.05) | 0.03 | 0.03 | 0.03 | 0.02 | 0.03 | 0.03 |
| G8 | 1 | 1.29 d | 1.08 d | 0.20 d | 0.16 c | 0.29 d | 1.80 d |
| | 2 | 1.37 c | 1.27 c | 0.26 c | 0.26 b | 0.37 c | 1.96 c |
| | 3 | 1.49 b | 1.32 b | 0.32 b | 0.24 b | 0.40 b | 2.01 b |
| | 4 | 1.64 a | 1.48 a | 0.41 a | 0.34 a | 0.44 a | 2.09 a |
| | LSD (0.05) | 0.02 | 0.03 | 0.02 | 0.03 | 0.02 | 0.03 |
| G9 | 1 | 1.09 d | 1.01 c | 0.14 d | 0.11 a | 0.22 c | 1.57 d |
| | 2 | 1.22 b | 1.15 a | 0.19 c | 0.13 a | 0.31 b | 1.77 c |
| | 3 | 1.27 b | 1.08 b | 0.24 b | 0.19 a | 0.33 b | 1.84 b |
| | 4 | 1.34 a | 1.76 a | 0.29 a | 0.26 a | 0.36 a | 1.92 a |
| | LSD (0.05) | 0.02 | 0.03 | 0.03 | 0.54 | 0.02 | 0.02 |
| G10 | 1 | 1.57 d | 1.39 d | 0.29 d | 0.25 d | 0.37 d | 1.87 d |
| | 2 | 1.68 c | 1.54 c | 0.36 c | 0.32 c | 0.44 c | 2.01 c |
| | 3 | 1.79 b | 1.64 b | 0.42 b | 0.35 b | 0.50 b | 2.09 b |
| | 4 | 1.86 a | 1.75 a | 0.46 a | 0.44 a | 0.56 a | 2.18 a |
| | LSD (0.05) | 0.03 | 0.03 | 0.02 | 0.02 | 0.02 | 0.04 |
| G11 | 1 | 1.38 d | 1.24 d | 0.26 d | 0.21 d | 0.32 d | 1.76 d |
| | 2 | 1.47 c | 1.4 c | 0.30 c | 0.26 c | 0.36 c | 1.87 c |
| | 3 | 1.65 b | 1.52 b | 0.37 b | 0.3 b | 0.52 b | 2.02 b |
| | 4 | 1.76 a | 1.64 a | 0.43 a | 0.37 a | 0.57 a | 2.09 a |
| | LSD (0.05) | 0.02 | 0.03 | 0.02 | 0.03 | 0.03 | 0.03 |
| G12 | 1 | 1.25 d | 1.11 d | 0.15 d | 0.12 a | 0.27 d | 1.74 d |
| | 2 | 1.31 c | 1.22 c | 0.21 c | 0.15 d | 0.33 c | 1.93 c |
| | 3 | 1.42 b | 1.25 b | 0.26 b | 0.21 c | 0.40 b | 2.02 b |
| | 4 | 1.54 a | 1.34 a | 0.32 a | 0.31 b | 0.44 a | 2.15 a |
| | LSD (0.05) | 0.02 | 0.03 | 0.03 | 0.03 | 0.03 | 0.02 |

Means in a column followed by the same latter are not significantly different at $p = 0.05$ using Duncan's new multiple range test (DNMRT); $T_1$ = 100% chemical fertilizer recommended rate (CFRR); $T_2$ = 5 t/ha chicken manure (CM), $T_3$ = 7 t/ha CM, and $T_4$ = 10 t/ha CM; NG = Nitrogen content in grain; NS = Nitrogen content in straw; PG = Phosphorus content in grain; PS = Phosphorus content in straw; KG = Potassium content in grain; KS = Potassium content in straw.

## 4. Discussion

All 20 of the investigated traits differed significantly amongst the 12 superior rice genotypes. These data reveal that all of these features in the genotypes have a lot of variation. Plant height is another important agronomic factor that influences rice plant yields indirectly. The tallest rice plants were observed with the treatment of 100% CFRR ($T_1$), which is statistically with the application of 10 t ha$^{-1}$ of chicken manure ($T_4$) because it provided sufficient N for the plant. The high N content of chicken manure, which influences the vegetative stage of plant growth, may play a significant role in plant height. N fertilizers play a significant role in rice vegetative growth. Plant height variation owing to nutrition sources was assumed to be caused by variations in the availability of essential nutrients. Chemical fertilizers deliver nutrients that are easily soluble in soil solutions and are hence readily available to plants. The availability of nutrients from organic sources is increased by microbial action and improved soil physical condition. These findings were supported by [29].

Tillering is a crucial part of rice growth enhancement, since it is an important feature for grain yield. Different fertilizer rates have an impact on rice plant tiller production. In the present study, maximum number of tillers genotype BRRI dhan75 (466 tillers m$^{-2}$) was produced with $T_4$, which was followed by $T_3$ and $T_1$. $T_2$ had a shortage of N and other key elements needed for tiller production, but the other treatments had enough to produce a high number of tillers. Rice plant productivity is largely determined by the number of panicles rather than the total number of tillers. Hence, we observed the maximum number of panicles (422 panicles m$^{-2}$) with $T_4$ (chicken manure 10 t ha$^{-1}$) which was at per with $T_3$. From this study it was observed that excessive application of inorganic fertilizer is not required to develop panicles. Organic sources provide more balanced nutrition to the plants, particularly micronutrients, which results in improved tiller affectivity in plants produced with chicken manure [30]. This result is partially supported by [31]. There has been an increase in the number of tillers using organic fertilizer due to the availability of enough nutrients that can be absorbed quickly by plants, but this cannot be separated from the influence of organic material that contains micronutrients in assisting the process of growth and nutrient absorption in an optimal and effective manner. Dhaliwal et al. [30] reported that rice growth and yield are affected by complete and balanced fertilizer because it can supplement and replenish nutrients that have been washed away or taken away by the current crop harvest. Organic fertilizers increased the availability of P in the flower development process, thus affecting the life of panicles. Uncertain P element in plants can boost flower creation. This is because fertilizer formulation aids in the repair of peat characteristics, allowing for greater nutrient availability and absorption by plants [30]. Dhaliwal et al. [30] states that the addition of organic matter to the soil has a stronger influence on enhancing the character of the ground and grain, requiring substantial amounts of P rather than just increasing the nutrients in the soil [32].

Panicle length has a significant impact on the amount of grain contained, and it helps to reduce the average percentage of empty grain. Organic fertilizer is extremely useful in giving the nitrogen nutrient that is needed by rice plants to promote plant growth such as filing grain by increasing the number of grain as well as the percentage drops in grain contains hollow effecting a growing weight of milled dry grain obtained [33]. Zhou et al. [34] reported that the length of the panicle is one factor that determines the outcome of the rice plant's, therefore the prospective outcomes acquired by longer panicles tend to be larger than those obtained by shorter panicles.

A substantial response to varied doses of treatments on yield parameters such as panicle length, viable grains, and grain weight was seen in this study. The results of this study demonstrated that applying 10 t ha$^{-1}$ of chicken manure can improve panicle production and growth. The highest quantity of grains per plant on genotype BRRI dhan75 (224 filled grain panicle$^{-1}$) followed by Putra2 (214 filled grain panicle$^{-1}$) and MR309 (208 filled grain panicle$^{-1}$), which were statistically superior to any other treatments. It could be related to increased grain formation, as well as rice growth aided by chicken manure.

T$_4$ also improved grain fertility, which was statistically distinct from other treatments. When compared to the other treatments, T$_4$ produced an 87% higher percentage of full grains. Only applying chicken manures resulted in a considerable loss of grain due to lower fertility [12]. It was caused by the organic manure's lower nutrient capacity which did not match the rice plant's requirements for producing fertile grains. There were also significant differences in the 1000-grain weight of genotype BRRI dhan39 (27.76 g), as affected by variation in chicken manures. Due to the severe scarcity of critical nutrients, 1000-grain weight is largely mediated by genetic potential, but in this case, it dropped dramatically with the application of organic manures, as well as with other treatments, and the plants failed to produce a bold grain. Increased yield qualities with manure application can improve soil structure, improve nutrient exchange, and sustain soil health, thus reigniting interest in organic farming [35]. Because chicken manure is high in nitrogen, phosphorous, potassium, and other vital nutrients. Hoque et al. [36] observed Chicken manure has a positive effect on rice yield contributing characters.

The ultimate reflection of the yield components is yield. The changes in yield attributes caused by varied amounts of manure treatment resulted in disparities in rice yield in this study. Different doses of organic manures had a substantial impact on grain yield, straw yield, and biological yield. From the study, we observed that, among the treatments, T$_4$ (chicken manure 10 t ha$^{-1}$) produced the highest grain yield (8.02 t ha$^{-1}$). The findings indicated that chicken dung had a higher nutritional quality and nutrient balance, resulting in the highest yield [37]. Schmidt and Knoblauch [38] also observed similar findings. In this study, straw yield (9.88 t ha$^{-1}$), as well as biological yield (17.36 t ha$^{-1}$) highest with the treatment T$_4$ (chicken manure 10 t ha$^{-1}$), which was identical to the yield produced by T$_3$ (chicken manure 7 t ha$^{-1}$). The treatment T$_4$ delivered the highest level of primary necessary elements required for plant vegetative growth, resulting in the largest straw production. Biological yield is the sum of grain yield (economic yield) and straw yield, and thus it was followed the trend like straw yield. These results were supported by [12]. Harvest indexes were also significantly varied with the treatments. In this study T$_4$ produced the highest harvest index (50.64%), which was followed by T$_3$ and T$_1$. This might be due to better grain yield with a corresponding better biological yield. The plots on T$_2$ (chicken manure 5 t ha$^{-1}$) gave the lowest harvest index. The increased harvest index with the treatments was related to a higher economic yield due to increased nutrient availability.

The nitrogen element is vital in stem elongation, which is caused by the cleavage, elongation, and expansion of new meristematic stems and leaves on causes the plant to grow taller [39]. Moreover, it is mentioned that administering organic materials rich in N can increase chlorophyll, which can further enhance photosynthate in plants, resulting in a higher accumulation of photosynthetic generated. The build-up of photosynthates causes cell growth and differentiation, which manifests itself as an increase in height or length, as well as an increase in the size and area of the leaves. Various research reports have proven that organic fertilizers have been shown in a number of studies to boost crop yield by improving the physical, chemical, and biological conditions of the soil, allowing plants to thrive [13]. Furthermore, organic fertilizer can release nutrients slowly during decomposition, resulting in lingering effects that are particularly valuable for future planning.

This study revealed that application of chicken manure 10 t ha$^{-1}$ had a higher ability of nutrient content. The stronger the genotype's ability to use nutrients is, the higher the nutritional content efficiency. A plant's appearance is the result of genetic and environmental interactions. Both are intertwined and have a mutual influence. If the environment is not favorable, even plants with good genetic makeup may not operate optimally, and vice versa [40]. By providing rice plants with the nutrients that they require, the manure improved the environment for them. Furthermore, the application of 10 t ha$^{-1}$ of chicken manure provided the highest grain yields; nevertheless, this was not statistically different from other soil health and nutrient availability treatments to enhance rice growth.

This is due to the fact that 10 t ha$^{-1}$ of chicken manure is sufficient to meet the nitrogen requirements of rice plants. The physical, chemical, and biological aspects of the soil improve with suitable dosages of chicken manure, which enhances the physical properties and structure of the soil. Because of the loose soil environment, increasing structure will allow plant roots to grow appropriately. If the soil's physical qualities are favorable, root development will be deeper and more expansive, allowing plants to absorb more nutrients and water, thus increasing plant output. This is because increasing the amount of fertilizer applied does not guarantee that the plants will grow faster or produce bigger yields, especially if the soil variables are less favorable [41].

Chicken manure is a common fertilizer used by farmers, and it has the ability to bind water, provide food, increase humus content, improve soil structure, and support balanced microorganism activity in the soil [42]. The use of inadequate mineral or organic fertilizers in soil deficient in native plant nutrients reduced rice performance considerably. On the other hand, surplus application promotes biological growth at the price of economic production. T$_4$ had the greatest levels of N, P, and K in grain and straw of all genotypes. These findings are partially similar to those of [38,43] who obtained higher contents of nutrient elements such as N, P, and K in rice by applying chicken manure. The crop performance improved significantly when chicken manure was applied of 10 t ha$^{-1}$ in this study. Any chicken manure rate greater than 10 t ha$^{-1}$ is likely to cause nutrient imbalance or toxicity, which could explain the poor results observed.

## 5. Conclusions

Based on the findings, it can be stated that organic fertilizer in the form of chicken manure has the ability to improve the growth parameters, yield, yield components, and nutrient content of rice grain and straw. The BRRI dhan75 genotype demonstrated positive benefits on rice growth, yield, and nutrient content when used an organic fertilizer. As a result, 10 t ha$^{-1}$ of chicken manure should be applied to maintain soil health and maximize output.

**Author Contributions:** Conceptualization, M.A. and M.Y.R.; data curation, M.A.; formal analysis, M.A.; funding acquisition, M.A. and M.Y.R.; investigation, S.I.R. and N.M.J.; methodology, M.A. and M.Y.R.; supervision, M.Y.R.; validation, M.Y.R.; writing—original draft, M.A.; writing—review and editing, M.A., M.F.I. and M.A.H. All authors have read and agreed to the published version of the manuscript.

**Funding:** Bangladesh Agriculture Research Council, Ministry of Agriculture, Bangladesh [PIU-BARC, NATP-2] and University Putra Malaysia-under the research grant ID-6282519.

**Institutional Review Board Statement:** Not applicable.

**Informed Consent Statement:** Not applicable.

**Data Availability Statement:** All data are included and explained in the manuscript text.

**Acknowledgments:** The authors are grateful to the Ministry of Agriculture (MoA), Bangladesh Agricultural Research Council (BARC-Project of NATP Phase-II), Bangladesh Rice Research Institute (BRRI) of the People's Republic of Bangladesh. Thanks also go to Universiti Putra Malaysia (UPM), Malaysia.

**Conflicts of Interest:** There is no conflict of interest.

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
