# Peer review of "The Nutrient Content, Growth, Yield, and Yield Attribute Traits of Rice (Oryza sativa L.) Genotypes as Influenced by Organic Fertilizer in Malaysia"

_sustainability, doi:10.3390/su14095692_

Round 1

Reviewer 1 Report

In this study, the authors conducted a two-season field experiment in 2020, using 12 rice genotypes. There were four treatments, i.e. chemical fertilizer, 5 t/ha, 7 t/ha, and 10 t/ha chicken manure. The authors analyzed physio-chemical parameters of soil, yield and yield components, photosynthesis rate, N, P, K contents of grain and straw. This is useful study to improve soil quality and the utilization of chicken manure. However, the nutrition contents of chicken manure is missing. There were mistakes in unit for example line 33, should be number of panicles m-2 (422.56). The English should be improved, there were grammar mistakes. The economic benefits for farmers should be given. The best recommendation from authors is   10 t/ha chicken manure. However, the maximum inputs of chicken manure is 10 t/ha, is it possible that the grain yield would increase further when extra chicken manure  was applied?

Reviewer 2 Report

This research can be of great interest for the worldwide research community.

Author Response

Please the attachment

Reviewer 3 Report

Comments

#1. please recheck the whole text for grammatical errors, try using correct tenses [past tense will be better since this is a finished study].

#2. The introduction is not sharp to the point it has lots of repetition in meaning and is not providing a logical flow of ideas leading to the problem statement. 

#3. The scientific problem and justification for this study need to be sharpened to clearly come out for the future readers and researchers that may make reference to this study.

#4. Please crosscheck the methodology and your results. the Methods question mentions only 3 treatments with 3 replications organized in a CRBD but in the results, another treatment (T4 =10t/ha application of Chicken manure) is not detailed at all in the methds. 

#5. In the statistical data analysis, the influence of interacting factors [interaction effects] are not taken care of, I advise that this be considered in the revision steps. Only after this is done, you can confidently attribute the observations to the impact of the treatments. 

#6. In the results section, the effect of genotype on the yield and growth parameters is not well discussed. if this does not clearly come out then the whole logic of studying different varieties is lost, so please authors let this come out clearly and more scientifically so. 

#7. The discussion should be structured for clarity not mixing up everything. The structuring should that a single paragraph or two discuss a single important component of the results and not mix within the same paragraph. because of the lack of structure, there are contradictions thorough the text for instance in Line 427 (pdf version) it says that treatment [T5=5t/ha CM] had the shortest plants and in Line 430, it says that " the different rates of fertilizers yielded identical outcomes". please crosscheck the whole discussion section and improve the results structure and presentation.

Round 2

Reviewer 1 Report

this revised version could be accepted.

Reviewer 3 Report

#1. The background could be condensed to make the academic problem addressed in this paper more precise and clear

#2. The logic of examining more than 5 genotypes is well linked with the problem, the discussion, and the conclusion. 

#3. The authors please to try to provide a scientific explanation of why chicken manure at 10t/ha was the best organic fertilizer application rate according to Table 3

#4.  The authors need to show how the delt with the interaction effects during the whole experimentation and then they are able to asribe growth in all measured parameters to the organic fertilizer (chicken manure application rate) effect

Author Response

Please the attachment
